# Combined Toxicity of the Most Common Indoor Aspergilli

**DOI:** 10.3390/pathogens12030459

**Published:** 2023-03-14

**Authors:** Daniela Jakšić, Dubravko Jelić, Nevenka Kopjar, Maja Šegvić Klarić

**Affiliations:** 1Department of Microbiology, Faculty of Pharmacy and Biochemistry, University of Zagreb, 10000 Zagreb, Croatia; 2Selvita, 10000 Zagreb, Croatia; 3Institute for Medical Research and Occupational Health, 10000 Zagreb, Croatia

**Keywords:** cytotoxicity, genotoxicity, proinflammatory cytokines, series *Nigri*, section *Flavi*, series *Versicolores*

## Abstract

The most common Aspergilli isolated from indoor air samples from occupied buildings and a grain mill were extracted and analyzed for their combined (*Flavi* + *Nigri*, *Versicolores* + *Nigri*) cytotoxic, genotoxic and pro-inflammatory properties on human adenocarcinoma cells (A549) and monocytic leukemia cells induced in macrophages (THP-1 macrophages). Metabolite mixtures from the Aspergilli series *Nigri* increase the cytotoxic and genotoxic potency of *Flavi* extracts in A549 cells suggesting additive and/or synergistic effects, while antagonizing the cytotoxic potency of *Versicolores* extracts in THP-1 macrophages and genotoxicity in A549 cells. All tested combinations significantly decreased IL-5 and IL-17, while IL-1β, TNF-α and IL-6 relative concentrations were increased. Exploring the toxicity of extracted Aspergilli deepens the understanding of intersections and interspecies differences in events of chronic exposure to their inhalable mycoparticles.

## 1. Introduction

Filamentous fungi are widely distributed in the environment as they quickly spread and reproduce by releasing airborne spores and fungal fragments. Possible health hazards arise from the risks of allergies or infection in a direct relation to the immunity of the exposed individuals in indoor environments [1]. Fungi produce different groups of volatile organic compounds and secondary metabolites. Despite a significant amount of research dedicated to investigation of indoor fungi [2,3,4,5,6], a clear mechanism by which these compounds contribute to the adverse health outcomes of exposed people has not yet been elucidated. Probably, the most notable studies related to indoor air quality were dedicated to “Sick Building Syndrome”, which includes respiratory, neurotoxic, chemosensory, dermatologic and other groups of symptoms observed in people who live and/or work in indoor environments of poor quality of air [7]. Special attention was granted to moldy dwellings in which *Aspergillus versicolor* and *Stachybotrys chartarum* were recognized as major contributors to observed respiratory symptoms [4,8,9,10].

Because of their biomedical and economical significance, special importance among indoor Aspergilli is given to those from the section *Flavi* and series *Nigri* (ex-section *Nigri*) and *Versicolores* (ex-section *Versicolores*), among which *A. flavus*, *A. tubingensis*, *A. welwitschiae*, *A. jensenii* and *A. creber* are the most commonly identified species [11,12,13,14,15,16,17]. The ability to tolerate a range of the water activities (a_w_) 0.75–0.90, a wide range of pH and ionic strength make *Aspergilli* particularly common indoor fungi [18]. They represent an important group of human and animal pathogens, but the worldwide significance of Aspergilli is due to their ability to produce mycotoxins such as aflatoxin B1 (AFB1) in the section *Flavi*, fumonisin B2 (FB2) in the series *Nigri* or sterigmatocystin (STC) in the series *Versicolores*. However, apart from these mycotoxins, Aspergilli produce thousands of different metabolites and there is limited information on their biological activity [19]. It was noted that exposure to fungal metabolites in the environment can promote health risks with fatal outcomes in sensitive predisposed individuals [20].

The infrageneric classification of the genus *Aspergillus* is difficult and complex and thus different approaches were suggested over time, especially through the development of sophisticated molecular and bioinformatic tools [21,22,23]. Brand new changes in species recognition and delimitations were introduced in the series *Versicolores* [24] and series *Nigri* [25] by the application of multispecies coalescence model-based methods, a novel phylogenetic approach. This resulted in *A. versicolor*, *A. creber*, *A. sydowii* and *A. subversicolor* as the only four species retaining validity in the series *Versicolores* [24] and *A. brasiliensis*, *A. eucalypticola*, *A. luchuensis*, *A. niger*, *A. tubingensis* and *A. vadensis* in the series *Nigri* [25]. However, besides their morphological properties and genetic variability, secondary metabolites are particularly useful in the classification and identification of fungi [26], as the chemoconsistency in the profiles of exometabolites among different isolates of a species was demonstrated [27,28]. The main purpose of the identification of fungi is to assign a certain biological activity to a certain species. Thus, it is useful to observe the activity of a full extrolite profile in appropriate in vitro models.

Indoor dust contains both indoor and outdoor fungi brought from various outdoor sources and thus it is the most representative sample for assessment of human exposure [29]. Aspergilli from the section *Flavi* and series *Nigri* and *Versicolores* have been frequently identified in indoor dust samples from around the world [13,30,31,32]. In addition, Aspergilli metabolites have also been identified in various indoor dust samples [9,33,34,35]. Therefore, the toxicological outcomes of the extracted Aspergilli, applied as a single and combined treatment on human cell lines, point to the possible outcomes on a cellular level following the inhalation of the metabolites produced by different species of *Aspergillus*. In this research, we focused on the section *Flavi*, the series *Nigri* and the series *Versicolores* because of their prevalence in indoor environments and the economic importance of this Aspergilli. To better understand interspecies and inter-series differences in the toxigenicity of medically and economically important Aspergilli, we combined the extracts of Aspergilli from *Flavi*, *Nigri* and *Versicolores* and compared their combined effects in vitro (*Flavi* + *Nigri* and *Versicolores* + *Nigri*) with the corresponding single treatment. Specifically, we combined AFB1-producing *A. flavus* with FB2-producing *A. welwitschiae* and *A. niger* and AFB1-nonproducing *A. flavus* with FB2-nonproducing *A. luchuensis*. Likewise, STC-producing Aspergilli from the series *Versicolores*, *A. creber* and *A. jensenii*, were selected for combinations with FB2-producing *A. welwitschiae* and *A. niger*. Additionally, to further explore the activity within the series *Versicolores* and *Nigri*, *A. protuberus* and *A. luchuensis* combinations were included in this research.

Inhalation is the most common route of exposure to indoor Aspergilli, so we performed MTT and comet assay to evaluate cytotoxicity and genotoxicity, respectively, in lung-derived adenocarcinoma cells (A549). First line protection upon the inhalation of particulate matter and associated chemical macrophages triggers the inflammatory response through the secretion of the cytokines. The THP-1 macrophages show similarities to peripheral blood mononuclear cells [36], so along with A549 cells THP-1 macrophages represent a good respiratory surrogate model for studying the toxicity expected following inhalation. To understand the possible effect on activity of the macrophages, using cell viability assay (MTT), relative concentrations of pro-inflammatory cytokines IL-1β, IL-5, IL-6, IL-8, IL-17 and TNF-α were measured in the cell medium upon the treatment of THP-1 macrophages.

## 2. Materials and Methods

### 2.1. Chemicals and Media

Acetonitrile, ethyl acetate, formic acid and methanol were purchased from Kefo (Sisak, Croatia); Czapek Yeast Agar (CYA) was purchased from Oxoid (Hampshire, UK). Medium RPMI 1640, fetal bovine serum (FBS), phosphate buffered saline (PBS; without Ca^2+^ and Mg^2+^), trypsin-EDTA, antibiotics penicillin and streptomycin were from Lonza (Basel, Switzerland). The MTT reagent [3-(4,5-dimethylthiazol-2-yl)-2,5-diphenyltetrazolium bromide], acridine orange, ethidium bromide, normal melting point agarose (NMP), low melting point agarose (LMP), phorbol 12-myristate 13-acetate (PMA), Triton X-100 and Tris buffer were purchased from Sigma-Aldrich (Deisenhofen, Germany) and molecular biology grade dimethyl sulfoxide (DMSO), used for dissolving the extracts and for the control treatment, was purchased from Sigma-Aldrich (Deisenhofen, Germany); technical grade DMSO, used for dissolving the formazane, ethanol, sodium chloride, disodium-EDTA and sodium hydroxide were from Kemika (Zagreb, Croatia).

### 2.2. Cytotoxicity, Genotoxicity and Immunomodulation of Combined Extracted Aspergilli

#### 2.2.1. Cell Culture and Treatment

Human lung adenocarcinoma cells A549 and human monocytic leukemia cells THP-1 (European Collection of Cell Cultures, Salisbury, UK) were grown at 37 °C in 5% CO_2_ in 75 cm^2^ flasks in RPMI supplemented with glutamine (2 mmol/L), heat-inactivated FBS 10%, penicillin (100 IU/mL; 1 IU 67.7 μg/mL) and streptomycin (100 μg/mL). Extracted Aspergilli were prepared as published previously [12,17,37] following the microextraction procedure according to Smedsgaard et al. [38]. Weighed dried fungal extracts were dissolved in 100% DMSO. The final concentrations of fungal extracts applied as the single treatment (Table 1), combination treatment (Table 2), as well as DMSO in the control treatments (≤0.1%), were obtained by dilution with the culture medium. Viability of the cells treated with the control containing DMSO was expressed as a percentage of viability produced in medium without DMSO. The concentration of each treatment was chosen based on the cytotoxicity screening assessed by MTT assay. The first criterion was that the concentrations of choice had non-significant effect on either A549 or THP-1 macrophage-like cell viability. Each extract was previously tested on most significant mycotoxins, i.e., AFB_1_ for the section *Flavi* [12], FB_2_ for the series *Nigri* [37] and STC for the series *Versicolores* [17]. Thus, the second criterion was to combine the mycotoxin producers from the section *Flavi* or the series *Versicolores* with the mycotoxin-producing isolates from the series *Nigri* and mycotoxin non-producing isolates from the sections *Flavi* or series *Versicolores* with the series *Nigri*. Even though this criterion was not fully met in the series *Versicolores*, *A. protuberus* was still included as it produced 100–200 times lower amounts of STC compared to selected producers of STC.

#### 2.2.2. MTT Proliferation Assay

The MTT proliferation test was used to test the cell viability of A549 cells and macrophage-like THP-1 cells as described previously [12,17,37]. The A549 (10^4^ cells/well) and macrophage-like THP-1 cells (5 × 10^4^ cells/well) were grown in a 96-well flat-bottom microplate in RPMI 1640 medium supplemented with 10% of FBS. The differentiation of THP-1 cells into macrophages was performed using PMA (40 nmol/L) directly in 96-well plate. Following 24 h treatment (Table 1 and Table 2), the medium was removed and 100 μL of MTT-tetrazolium salt reagent diluted in RPMI without FBS (0.5 mg/mL) was added into each well and incubated for 3 h. Afterwards, the medium was replaced with DMSO (100 µL) to dissolve formazan, a product of metabolized MTT reagent. The cells were incubated at room temperature on a rotary shaker for 15 min and the absorbances were measured at a wavelength of 540 nm (Labsystem iEMS, type 1404, Berthold Technologies GmbH & Co.KG, Bad Wildbad, Germany). All tests were performed in at least three replicates and results were expressed as percentage of control.

#### 2.2.3. Alkaline Comet Assay

A549 cells (3 × 10^5^ cells per well) were seeded in 6-well plates for 24 h and then the cell medium was replaced with the treatment or control medium (Table 1 and Table 2) for 24 h. The comet assay was carried out according to Singh et al. (1988) [39] with minor modifications as published previously [12]. Following the 24 h treatment (Table 1 and Table 2), the cells were washed with cold PBS, scraped with rubber and resuspended in PBS. Aliquots (20 μL) of this suspension were mixed with 100 μL 0.5% LMP (in Ca-and Mg-free PBS) and 100 μL of agarose-cell suspension was spread onto a fully frosted slide pre-coated with NMP agarose (1% in sterile distilled water). The slides were solidified on ice for 10 min and submerged in the alkaline lysis solution [39] at +4 °C. After one hour, the slides were placed in denaturation and electrophoresis buffer (10 mmol/L NaOH, 200 mmol/L Na_2_EDTA, pH 13), incubated for 20 min and electrophoresed for 20 min at 25 V and 300 mA. Neutralization solution (0.4 mol/L Tris/HCl, pH 7.5) was applied to each microgel. The slides were kept in a humid atmosphere in a dark box at +4 °C until further analysis. The DNA was stained with 100–250 μL ethidium bromide solution (20 μg/mL) per slide for 10 min and observed under fluorescence microscope (Olympus, Tokyo, Japan). The slides were scored using an image analysis system (Comet assay IV, Instem-Perceptive instruments Ltd., Stone, Staffordshire, UK). All experiments were performed in duplicate. Images of 200 randomly selected cells (100 cells from each of the two replicate slides) were measured and percentage of DNA in the comet tail (or tail intensity) [40] was taken as a reliable measure of genotoxicity.

#### 2.2.4. Determination of Cytokine Levels

THP-1 cells were seeded (1 × 10^6^ cells per well) on 24-well cell culture plates, differentiated for 24 h with phorbol myristate acetate (PMA; 40 nmol/L. Following the 24 h treatment with extracted Aspergilli and the control (Table 1 and Table 2), the cell supernatant was harvested and frozen at −80 °C until analysis. Concentrations of TNF-α, IL-1β, IL-6, IL-5, IL-8 and IL-17 were determined in the cell supernatant by DuoSet ELISA kits (R&D systems, Minneapolis, MN, USA) following the instructions provided by the manufacturer and as described elsewhere [12]. Cytokine concentrations were calculated from measured absorbances at 450 nm (En Vision^®^ Multilabel Plate Reader, Perkin Elmer, Waltham, MA, USA) and by standard calibration curves and results of three replicates were expressed as percentage of control.

## 3. Statistics

In the MTT test and comet assay, the results were presented as arithmetic mean ± standard deviation (SD). The relative cytokine concentrations were presented as mean ± standard error of mean (SEM). The normality of data distribution was tested by the Kolmogorov–Smirnov test, one-way-ANOVA was applied for normally distributed data followed by the Sidak post-test, while for non-normally distributed data the Kruskal–Wallis test was applied followed by Dunn’s multiple comparison test. *p* < 0.05 was considered statistically significant for all calculations.

## 4. Results

### 4.1. Cytotoxicity

The cytotoxicity of the extracted *A. flavus* (AFB1-producing and non-producing), *A. niger*, *A. welwitschiae*, *A. creber* and *A. jensenii* applied as a single treatment in A549 cells and THP-1 macrophages was previously published [12,17,37]. A549 cells were more sensitive to the combination of *Versicolores* + *Nigri* than *Flavi* + *Nigri* (Figure 1a,b). When combined with *A. niger* and *A. welwitschiae*, only higher concentration of AFB1-producing *A. flavus* (0.2 mg/mL) significantly reduced A549 cell viability and the combination with *A. niger* had a more pronounced effect on the viability drop (Figure 1a). The most pronounced effect on decrease in the cell viability had combinations of *A. jensenii* and *A. creber* with *A. niger* whereas a 15% higher impact was achieved by *A. jensenii* (Figure 1b). The combinations with *A. welwitschiae* resulted in a significant decrease in the cell viability only when a higher concentration of *A. jensenii* (0.05 mg/mL) and lower concentration of *A. creber* (0.006 mg/mL) was combined. Thus, *A. jensenii* is more potent than *A. creber* when combined with *A. niger*, and oppositely, *A. creber* is more potent than *A. jensenii* when combined with *A. welwitschiae*.

The combination *A. protuberus* + *A. luchuensis* was less potent than the corresponding single extract treatment (Figure 1a,b, Appendix A).

The THP-1 macrophages were less sensitive to all combinations of extracted Aspergilli compared to A549 cells (Figure 2). The lower concentrations of AFB1-producing *A. flavus* (0.1 mg/mL) combined with *A. niger* or *A. welwitschiae* decreased the viability of the THP-1 macrophages whereas the combination of AFB1-nonproducing *A. flavus* (0.05 mg/mL) with *A. luchuensis* slightly increased the cell viability (Figure 2a). The combinations of *Versicolores* with *Nigri* had a lower impact on the viability of THP-1 macrophages compared to single treatment (Figure 2b). A pronounced proliferative effect was observed when a lower concentration of *A. jensenii* extract (0.005 mg/mL) was combined with *A. niger* or *A. welwitschiae* (Figure 2b).

### 4.2. Genotoxicity

The genotoxicity of the applied treatment was evaluated through the percentage of the DNA in the comet tail. The genotoxicity of extracted *A. flavus* (AFB1-producing and non-producing), *A. niger* and *A. welwitschiae* applied as a single treatment was previously published [12,37]. The DNA damage produced by all the single isolates was insignificant, except for *A. welwitschiae* extract. All combinations of *Flavi* and *Nigri* had a more pronounced effect on DNA damage compared to the corresponding single treatment except for the tail intensity reported for the AF1a,b + AN2 combination, which was not higher than those observed for the single treatment AN2 (Figure 3a). The most pronounced increase in the tail intensity was produced when a lower concentration of AFB1-producing *A. flavus* (0.1 mg/mL) was combined with *A. niger* but the genotoxicity dropped for a higher concentration of *A. flavus* in the same combination (Figure 3a). The combination of AFB1-producing *A. flavus* with *A. welwitschiae* showed a significant increase in tail intensity in a concentration-dependent manner; however, the impact of a single treatment with *A. welwitschiae* was cancelled by the addition of AFB1-producing-*A. flavus*. Additionally, it was up to four times lower compared to the combination of AFB1-producing *A. flavus* with *A. niger*. A significant increase in tail intensity was also achieved when AFB1-nonproducing *A. flavus* was combined with *A. luchuensis*, but only when a lower concentration of *A. flavus* was applied. All extracts assigned to the series *Versicolores* produced significant DNA damage when applied as a single treatment (*A. jensenii* > *A. creber* > *A. protuberus*) (Figure 3a,b). The genotoxicity of *A. jensenii* and *A. creber* decreased in combinations with *Nigri*; however, the impact on produced tail intensities remained significant for the combinations with *A. creber*. The impact on the DNA damage following the treatment with *A. protuberis* was not significantly changed when applied in combination with *A. luchuensis* (Figure 3b).

### 4.3. Cytokines IL-1β, IL-5, IL-6, IL-8, IL-17, TNF-α Production

Previously, there were reported impacts of Aspergilli from the series *Nigri* on TNF-α, IL-1β, IL-6, IL-8 and *Flavi* on TNF-α, IL-1β, IL-6, IL-8, IL-17 cytokine production in THP-1 macrophages [37,39]. To provide a uniformity in data presentation in this research, we provide the results of IL-1β, IL-5, IL-6, IL-8, IL-17, TNF-α excretion by THP-1 macrophages for extracted Aspergilli from the sections *Flavi*, *Nigri* and *Versicolores*, single and combined *Flavi* + *Nigri* (Figure 4a) and *Versicolores* + *Nigri* (Figure 4b).

All the extracted Aspergilli applied as a single treatment or combined had a significant impact on the decrease in relative concentration of IL-5 compared to control (Figure 4). The relative concentrations of IL-1β, TNF-α and IL-6 significantly increased and IL-17 significantly decreased when AFB1-producing *A. flavus* was combined with *A. niger* or *A. welwitschiae* (Figure 4a). The opposite was observed when AFB1-nonproducing *A. flavus* was combined with *A. luchuensis*. While the concentrations of IL-1β and TNF-α increased compared to *A. luchuensis*, the relative concentration of IL-17 was significantly lower compared to a single treatment with AFB1-nonproducing *A. flavus* and *A. luchuensis*, especially when a higher concentration of AFB1-nonproducing *A. flavus* (0.1 mg/mL) was applied.

Single *Versicolores* had the most pronounced impact on IL-17 (Figure 4b). When *A. jensenii* or *A. creber* were combined with *A. niger* or *A. welwitschiae*, relative concentrations of TNF-α were similar to those provoked by *A. niger* or *A. welwitschiae* single treatments, Figure 4b. The combination of *A. protuberus* + *A. luchuensis* significantly impacted an increase in relative concentrations of IL-6 and IL-1β but this effect was less pronounced when a higher concentration of *A. protuberus* (0.09 mg/mL) was applied (Figure 4b).

## 5. Discussion

In the previously published research, the emphasis was given to different extracts from each of the sections/series in which the significance of the mycotoxins STC [17], FB2 [37] and AFB1 [12] for the toxicity of Aspergilli from the sections *Versicolores*, *Nigri* and *Flavi*, respectively, was observed. The purpose of the research presented in this manuscript was to point to the toxicological outcomes expected following exposure to the mixture of extrolites produced by important indoor Aspergilli belonging to the section *Flavi*, the series *Nigri* and the series *Versicolores*. Specifically, the impact on viability and the DNA damage in A549 cells, as well as the impact on the viability of the THP-1 macrophages and the excretion of the proinflammatory cytokines presented.

Different results were obtained when *Nigri* were combined with *Flavi* and *Versicolores.* The combination *Flavi* + *Nigri* dominated the cytotoxic and genotoxic impact produced by the combination with *A. niger*. When the combination *Versicolores* + *Nigri* was applied in A549 cells, a significant decrease in cell viability was achieved when *A. jensenii* was combined with *A. niger*, and *A. creber* with *A. welwitschiae*. Both *A. niger* and *A. welwitschiae* lowered the impact on the DNA damage produced by *A. creber* and *A. jensenii*. When *A. luchuensis* was combined with AFB1-nonproducing *A. flavus* and *A. protuberus*, the resulting impact was an increase in the viability of A549 cells; however, a significant impact on the DNA damage was observed, especially when *A. luchuensis* was combined with lower concentrations of AFB1-nonproducing *A. flavus* or *A. protuberus*. Both combinations of AFB1-producing *A. flavus*, with *A. niger* or *A. welwitschiae*, slightly decreased the viability of THP-1 macrophages but only when a lower concentration of *A. flavus* was applied. On the other hand, a pronounced proliferative effect was observed for the combinations with *A. jensenii*. The combination of *A. luchuensis* with AFB1-nonproducing *A. flavus* and *A. protuberus* resulted in an increase in cell viability compared to the single treatment.

The conducted research also points to some interspecies and interseries differences in the cytotoxicity, genotoxicity and proinflammatory effects of Aspergilli from the section *Flavi* and the series *Nigri* and *Versicolores*. The impact of the combinations (*Flavi* + *Nigri* and *Versicolores* + *Nigri*) was compared to a single treatment. While the concentrations of each extract applied as a single treatment were intended to be uniform, there are some differences in their concentrations as is shown in Table 1. For example, applied concentrations of AFB1-producing and nonproducing *A. flavus* are different, 0.1 and 0.2 mg/mL vs. 0.05 and 0.1 mg/mL, because the extract of AFB1-producing *A. flavus* was adjusted to provide a better insight into a possible role of AFB1 in the assessed toxicity of the extract as presented and discussed previously [12]. Similarly, the assessed concentrations of *A. jensenii* and *A. creber* are much lower as they were adjusted to the content of STC [17]. Additionally, subcytotoxic concentrations are advised to be tested for genotoxicity by comet assay [40].

Considering the concentration vs. observed biological effect of each extract, generally we can conclude that the cytotoxic potency of *Versicolores* is more than five times higher compared to *Flavi* and two to six times higher than *Nigri.* The THP-1 macrophages are even more sensitive than A549 cells, where *Versicolores* are six to twelve times more potent than *Flavi* and two to sixteen times more potent than *Nigri*. The extract of *A. protuberus*, which contained an inconsiderable amount of STC, had a 25–30% higher impact on cell viability than the extracts of *A. creber* and *A. jensenii*, which contained 500 times higher amount of STC. Similarly, decreased viabilities measured after exposure to FB2 non-producing *A. luchuensis* were 30% lower compared to FB2-producing *A. welwitschiae.* This evidently shows that Aspergilli are the source of toxic secondary metabolites apart from the recognized mycotoxins. When applied simultaneously, many mycotoxins exert complex interactions, i.e., additive or synergistic in one concentration range and antagonizing in another [41]; even more complex interactions are expected if the simultaneous presence of several thousand different metabolites in the mixture is considered.

Even though STC is the dominant biologically active metabolite in the selected extracts of the *Versicolores*, the presence of other metabolites can modulate its toxic effects [17]. Recent research revealed that STC, 5-methoxysterigmatocystin (5-MET), versicolorins, tryprostatin B, hydroxysidonic acid and notoamides are the prevailing metabolites in indoor environments produced by Aspergilli, namely by Aspergilli from the series *Versicolores* [33]. Although STC is the main metabolite of *Versicolores*, other secondary metabolites may contribute to observed biological outcomes. A group of prenylated indole alkaloids-notoamides may be produced by some *Verscolores*, including *A. protuberus* [42]. A moderate cytotoxic potency of some notoamides was shown against immortalized cervical cancer cells (HeLa) and lymphoblast-like cells (L1210) with IC_50_ values in the range of 0.022–0.052 mg/mL [43]. This group of metabolites may be implicated in the observed decrease in the viability of A549 cells and THP-1 macrophages following treatment with *A. protuberus*. Other studies conducted on A549 cells and THP-1 monocytes reported the high toxicity of STC-producing Aspergilli [44,45] and the genotoxicity of extracted *A. versicolor* [46]. Interestingly, decreased viabilities measured after exposure to FB2 non-producing *A. luchuensis* were 30% lower compared to FB2-producing *A. welwitschiae*.

The higher toxicity of AFB1-producing *A. flavus* compared to the non-aflatoxinogenic strain of *A. flavus*, but also compared to AFB1 [12] indicates a contribution of other metabolites to observed impacts on cell viability. Apart from AFB1, several secondary metabolites are recognized for their biological activity in the section *Flavi*, such as, mycotoxin cyclopiazonic acid, insecticides aflavinines, aflatrems and aflavazole or antifungal asperfuran [14]. These, but also many other yet uninvestigated metabolites, may contribute to the toxicity in human cell lines. The toxic properties of Aspergilli from the section *Nigri* cannot be assigned to FB2. However, the pronounced toxic effects of *A. welwitschiae*, namely in A549 cells, are of particular concern because of a high prevalence of this species in indoor environment [13,15,37]. When applied as binary combinations, *Flavi* + *Nigri* and *Versicolores* + *Nigri* exhibit significantly higher cytotoxicity compared to single species extracts, suggesting possible additive or synergistic cytotoxic effects in A549 cells. The binary combinations of *Flavi* + *Nigri* decreased the viability of THP-1 macrophages and increased genotoxicity in A549 cells equally or in a more pronounced manner than single species extracts, suggesting at least an additive effect. Opposite to that, dual combinations of the *Versicolores* + *Nigri* extracts evoked markedly lower effects on the viability of THP-1 macrophages and DNA damage in A549 cells compared to single species extracts, suggesting antagonism.

In the latest changes in taxonomy, *A. welwitschiae* is synonymized with *A. niger* and *A. luchuensis* with *A. piperis* [25]. In our previous research, we demonstrated a significant difference in the observed cytotoxic and genotoxic potency of the extracts of *A. luchuensis*, *A. piperis*, *A. welwitschiae* and *A. niger* [37]. The differences in biological impact suggest a different qualitative composition of the extracts assigned to *A. welwitschiae* vs. *A. niger* and *A. luchuensis* vs. *A. piperis*, which is in line with the chemoconsistency of the species. However, if *A. pipersis* is the same species as *A. luchuensis* and *A. welwitschiae* the same as *A. niger*, it is possible that the quantities of the same metabolites differ in different extracts, which reflects on the observed differences in biological activity.

All single and combined treatments significantly decreased the concentration of IL-5. Applied as a single treatment, Aspergilli provoked an increase in IL-17 concentrations; however, all combinations (*Flavi* + *Nigri* and *Versicolores* + *Nigri*) significantly impacted a decrease in IL-17. Additionally, all the combinations provoked an increase in IL-1β, TNF-α and IL-6 relative concentrations. Cytokines have an important role in the regulation of the host defense mechanisms in the events of inflammation and the tissue damage associated with infection, malignancies, autoimmune disorders or allergies. Most of them show pleiotropic activity, so it is difficult to drive a singular conclusion from their expression patterns following the treatment with extracted Aspergilli. TNF-α is an important endogenous pyrogen and immunoregulatory cytokine in charge of the production of IL-1, IL-6 and IL-8. Its deficiency in experimental animals is assigned to cancer promotion [47]. An increase in TNF-α is associated with enhanced phagocytosis as a response of polymorphonuclear leukocytes to *Aspergillus* hyphae [48]. Even though IL-1β has a beneficial role of mediating an immune response against pathogenic infiltration, it is a potent pro-inflammatory cytokine secreted by macrophages and monocytes that may be implicated in chronic inflammation because of promoted tissue damage [49,50]. The cytokines IL-1 and TNF-α act as regulators of IL-6 and IL-8 [51,52]. Even though IL-6 may also act as an anti-inflammatory, under circumstances of chronic inflammation, as in the case of chronic exposure to Aspergilli, it is rather pro-inflammatory. The cytokine IL-5 regulates the expression of genes involved in proliferation, cell survival and maturation and the effector functions of B cells, the most potent activator of eosinophils [53,54] and facilitator of T_H_2 cell-mediated pathologic responses [55]. Even though it is predominantly a T cell-derived cytokine, its secretion upon the stimulation of alveolar macrophages was demonstrated [55]. The relevance of an observed significant decrease in IL-5 relative concentrations by Aspergilli extracts is yet to be elucidated. While inflammation generally correlates with IL-5 overexpression, a lack of a functional gene for the IL-5 or the IL-5 receptor in mice corresponds to developmental and functional impairments in B-cell and eosinophil lineages [54]. Pro-inflammatory cytokine IL-17 is essential in the host’s defense against microbial infections, autoimmune diseases, metabolic disorders and cancer [56]. It may have a crucial role in allergic airway inflammation [57] and fungal infection, including *A. fumigatus*-caused aspergillosis [58]. In the experiment on mice exposed to conidia of *A. fumigatus*, which also contains a mixture of different metabolites, the extravasation of eosinophils from the blood into the lungs was mediated by IL-17 [59]. The previously reported cytotoxicity of Aspergilli from the series *Versicolores* in THP-1 [17] may be mediated by an increase in IL-17 relative concentrations. It could also explain the increase in cell viability following the combined treatment *Versicolores* + *Nigri* where a decrease in IL-17 is observed.

The presented research should be cautiously extrapolated to real-life exposure since short-term treatment experiments performed in cell cultures cannot predict the full impact of a continuous and/or prolonged exposure to the toxic compound(s). In addition, the production of secondary metabolites is a highly variable process depending on the substrate, host and environment [60,61] while the growth conditions of *Aspergilli* presented in the manuscript are considered the most suitable when a high-end production of secondary metabolites is to be achieved [62]. However, it is a good starting point when evaluating the significance of exposure to a certain fungal species. Some available data suggest that exposure to mycotoxins by the inhalation of the fungal spores in reality results in extremely low exposure to mycotoxins, e.g., the calculated dose for exposure to AFB1 is only 19 ng/kg/day [63]. However, other research showed that fungal fragments may be released in up to 320 times higher amounts than the spores [64], which drastically increases the quantities of the mycotoxins and other fungal metabolites to which people may be exposed. House dust is particularly loaded with fungal metabolites [31,33,65,66] and is considered the main route of exposure to fungal metabolites in indoor environments [67]. In events such as water leaks, flooding or building damage where lots of dust is produced and concentrations of fungal producers increase, people may be exposed to even larger quantities of different fungal metabolites. All of this may be considered and applied into the design of a better model for studying inhalation exposure effects. This could include qualitatively and quantitatively a well-defined mixture of the metabolites found in indoor dust or other inhalable particulate matter. In addition, a more advanced system than the cell lines should be used. Some of these advanced systems include physiologically relevant models of human organs, such as organs-on-chips [68]. Once the challenges regarding the architectural complexity of the human tissues and organs in vitro are resolved, such systems should be implemented in toxicity studies as a replacement for in vivo studies in animals.

## 6. Conclusions

Exploring the toxicity of mixtures of *Aspergilli* brings us closer to estimating the total impact of epithelial damage on airways and the activity of macrophages in events of chronic exposure to *Aspergilli*, namely their metabolites, which can persist in indoor environments for extended periods of time. However, to explain the mechanism behind observed effects, a thorough chemical analysis of the extracts is necessary as well as a better understanding of the complex physicochemical interactions occurring on a molecular level. Nonetheless, it is important to note that a different composition of the metabolites is expected under different conditions including media/matrix and the related availability of nutrients, microelements, pH and a_w_. Additionally, it is important to consider that different temperature conditions, levels of carbon dioxide and oxygen concentrations may also affect the secondary metabolism of fungi and reflect on toxicological outcomes. However, the obtained results point to some important intersections/interseries and interspecies differences among *Aspergilli*, especially in the series *Versicolores* and *Nigri*. The observed effects are the complex outcome of interactions of thousands of metabolites in each set of binary mixtures. The presented results also suggest that the metabolites from the series *Nigri* increase the cytotoxic and genotoxic potency of *Flavi* metabolites in A549 cells, while antagonizing the cytotoxic potency of *Versicolores* in THP-1 macrophages and genotoxicity in A549 cells. When combined, these Aspergilli have a more pronounced impact on cytokine excretion by THP-1 macrophages than the same Aspergilli applied as a single treatment. Accordingly, a more significant impact on immunomodulation is expected following exposure to complex extrolite mixtures.

## Figures and Tables

**Figure 1 pathogens-12-00459-f001:**
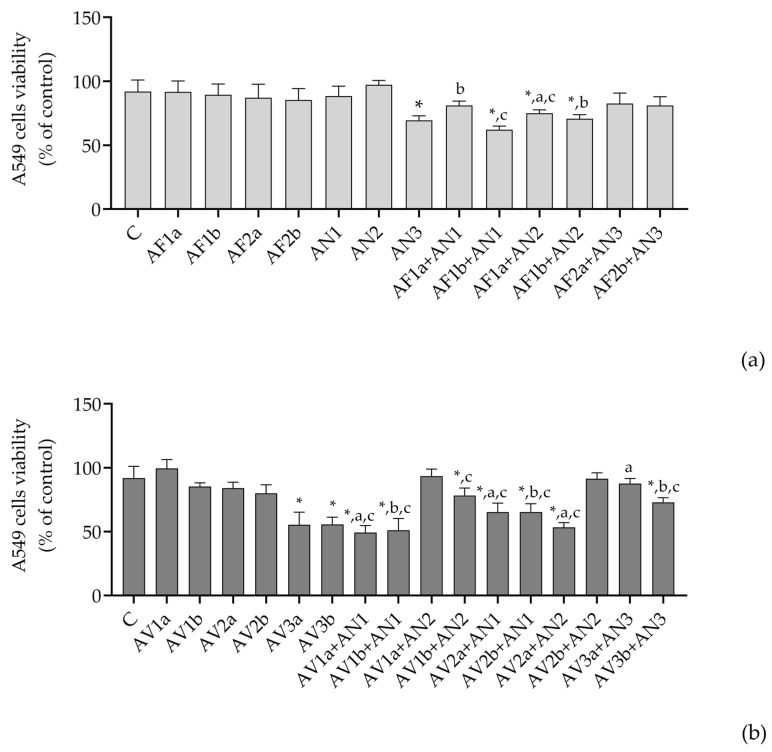
Cytotoxicity of A549 cells upon the treatment with the extracted Aspergilli from the section *Flavi* and the series *Nigri* (**a**) and the series *Versicolores* and *Nigri* (**b**), single and mixed as specified. The viability of the cells (% control treatment) is presented as mean and standard deviation derived from the MTT assay performed in triplicate at least. *—significantly different compared to control treatment (C), a—significantly different compared to AF1a or AF2a (**a**) and AV1a, AV2a or AV3a (**b**), b—significantly different compared to AF1b or AF2b (**a**) and AV1b, AV2b or AV3b (**b**), c—significantly different compared to AN1, AN2 or AN3 (**a**,**b**). *^,a,b,c^ *p* < 0.05 was considered statistically significant for all calculations. Viability of A549 cells following the control treatment (DMSO-containing) was 91.91 ± 9.154% of viability produced in DMSO-free medium.

**Figure 2 pathogens-12-00459-f002:**
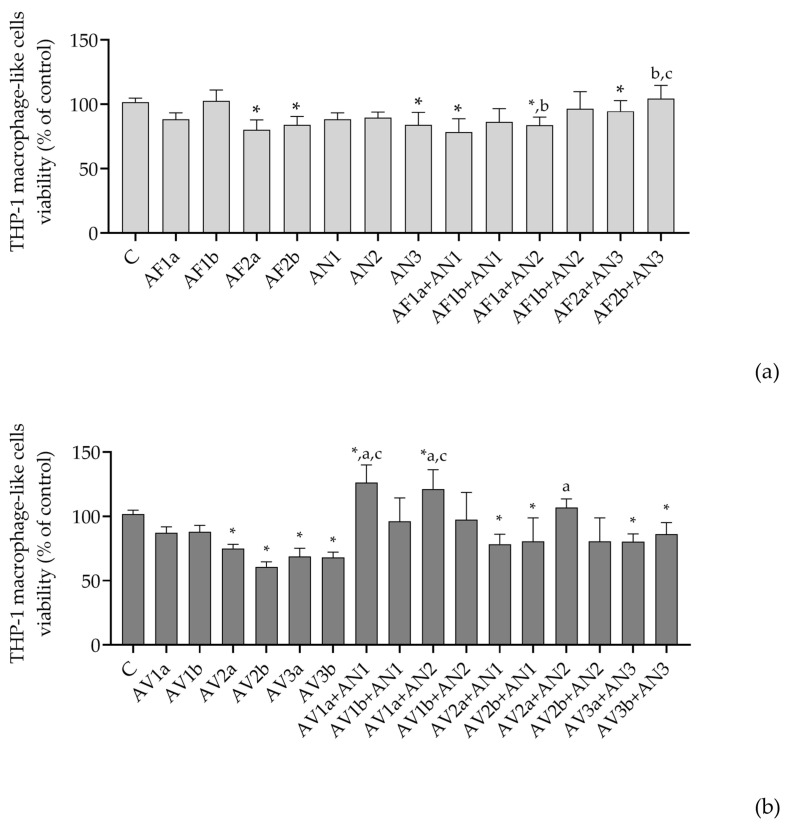
Cytotoxicity of THP-1 macrophage-like cells upon the treatment with the extracted Aspergilli from the sections *Flavi* and the series *Nigri* (**a**) and the series *Versicolores* and *Nigri* (**b**), single and mixed as specified. The viability of the cells (% control treatment) is presented as mean and standard deviation derived from the MTT assay performed in triplicate at least. *—significantly different compared to control treatment (C), a—significantly different compared to AF1a or AF2a (**a**) and AV1a, AV2a or AV3a (**b**), b—significantly different compared to AF1b or AF2b (**a**) and AV1b, AV2b or AV3b (**b**), c—significantly different compared to AN1, AN2 or AN3 (**a**,**b**). *^,a,b,c^ *p* < 0.05 was considered statistically significant for all calculations. Viability of THP-1 macrophages following the control treatment (DMSO-containing) was 101.6 ± 3.088% of viability produced in DMSO-free medium.

**Figure 3 pathogens-12-00459-f003:**
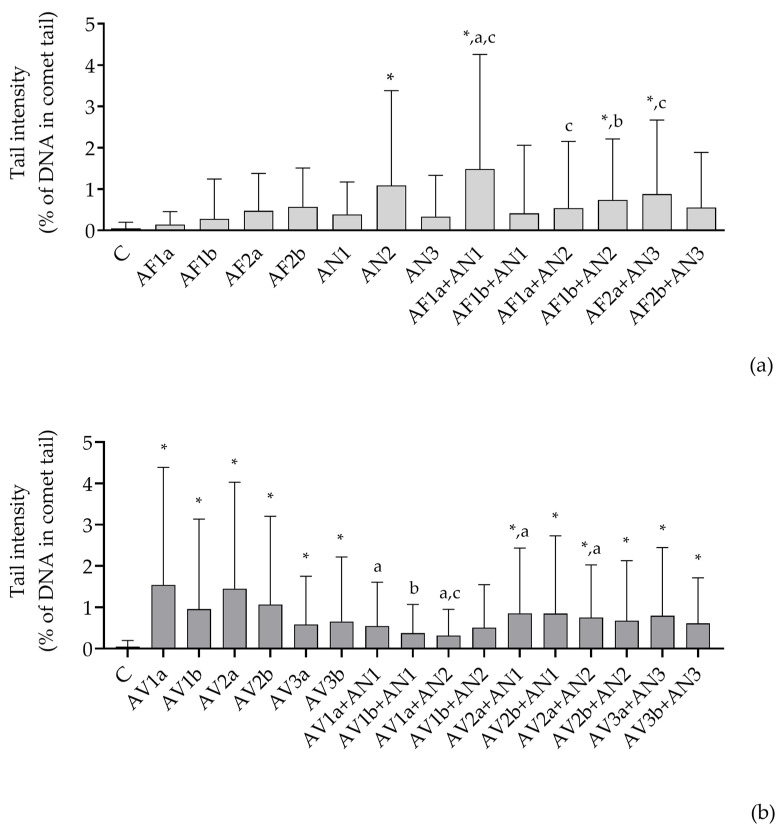
Genotoxicity of A549 cells upon the treatment with the extracted Aspergilli from the section *Flavi* and the series *Nigri* (**a**) and the series *Versicolores* and *Nigri* (**b**), single and mixed as specified. The DNA damage is evaluated through percentage of the DNA in comet tail and presented as mean ± SD derived from 200 measured cells. Each comet assay was performed in duplicate. *—significantly different compared to control treatment (C), a—significantly different compared to AF1a or AF2a (**a**) and AV1a, AV2a or AFVa (**b**), b—significantly different compared to AF1b or AF2b (**a**) and AV1b, AV2b or AV3b (**b**), c—significantly different compared to AN1, AN2 or AN3 (**a**,**b**). *^,a,b,c^
*p* < 0.05 was considered statistically significant for all calculations.

**Figure 4 pathogens-12-00459-f004:**
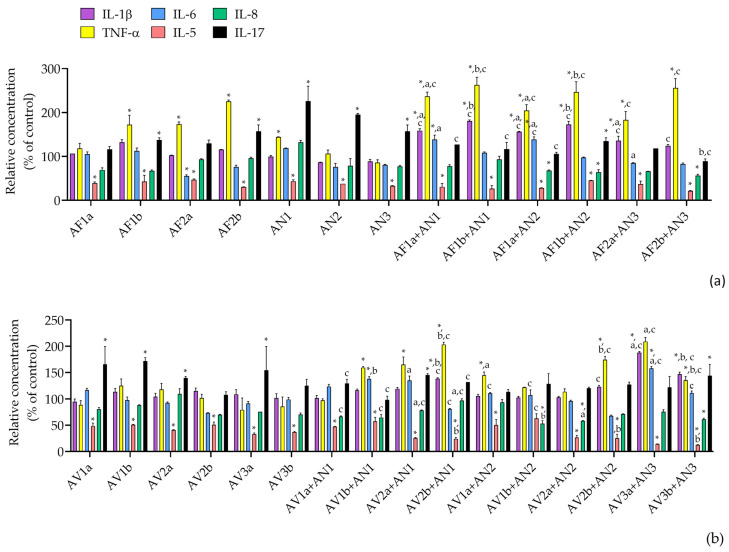
Relative concentration of the cytokines measured in the supernatant of THP-1 macrophage-like cells upon treatment with the extracted Aspergilli from the section *Flavi* and the series *Nigri* (**a**) and the series *Versicolores* and *Nigri* (**b**), applied as a single or combined treatment and compared to the control treatment. Values are presented as arithmetic mean and standard error of the mean. *—significantly different compared to control treatment (C), a—significantly different compared to AF1a or AF2a (**a**) and AV1a, AV2a or AV3a (**b**), b—significantly different compared to AF1b or AF2b (**a**) and AV1b, AV2b or AV3b (**b**), c—significantly different compared to AN1, AN2 or AN3 (**a**,**b**). *^,a,b,c^
*p* < 0.05 was considered statistically significant for all calculations.

**Table 1 pathogens-12-00459-t001:** Description of the treatment of A549 and THP-1 macrophage-like cells. The code assigned to control and the extracted Aspergilli is used elsewhere in the manuscript.

	Treatment Description	Short Code	Reference
Control	0.1% DMSO in RPMI medium	C	
section *Flavi*	*A. flavus* 0.1 mg/mL (0.75 µM AFB_1_)	AF1a	[12]
	*A. flavus* 0.2 mg/mL (1.5 µM AFB_1_)	AF1b	
	*A. flavus* 0.05 mg/mL (AFB_1_ negative)	AF2a	
	*A. flavus* 0.1 mg/mL (AFB_1_ negative)	AF2b	
series *Nigri*	*A. niger* 0.1 mg/mL (0.00009 µM FB_2_)	AN1	[37]
	*A. welwitschiae* 0.1 mg/mL (0.095 µM FB_2_)	AN2	
	*A. luchuensis* 0.1 mg/mL (FB_2_ negative)	AN3	
series *Versicolores*	*A. jensenii* 0.005 mg/mL (0.1 µM STC)	AV1a	[17]
	*A. jensenii* 0.05 mg/mL (1 µM STC)	AV1b	
	*A. creber* 0.006 mg/mL (0.1 µM STC)	AV2a	
	*A. creber* 0.06 mg/mL (1 µM STC)	AV2b	
	*A. protuberus* 0.05 mg/mL (0.0016 µM STC)	AV3a	
	*A. protuberus* 0.1 mg/mL (0.0024 µM STC)	AV3b	

**Table 2 pathogens-12-00459-t002:** The combinations of the extracted Aspergilli used in the treatment of A549 and THP-1 macrophage-like cells. The codes assigned to the treatment are used elsewhere in the manuscript.

	Abbreviations for Each Combination
*Flavi* + *Nigri*	AF1a + AN1
	AF1b + AN1
	AF1a + AN2
	AF1b + AN2
	AF2a + AN3
	AF2b + AN3
*Versicolores* + *Nigri*	AV1a + AN1
	AV1b + AN1
	AV1a + AN2
	AV1b + AN2
	AV2a + AN1
	AV2b + AN1
	AV2a + AN2
	AV2b + AN2
	AV3a + AN3
	AV3b + AN3

## Data Availability

Data is contained within the article and Appendix A. For any additional information please contact the corresponding author.

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
