# Peer review of "Combined Toxicity of the Most Common Indoor Aspergilli"

_pathogens, 2023, doi:10.3390/pathogens12030459_

Round 1
Reviewer 1 Report
· Page 2, § 2.1. Chemicals and media : DMSO appears twice, from two different suppliers
· Page 3, § 2.2.2. MTT Proliferation Assay : “the differentiation of THP-1 cells into macrophages was performed using PMA (40 nmol/l)”. It is not clear when this differentiation was performed. Is it directly in the 96 wells plate or in the 75 cm2 flask culture, i.e. that requires a trypsinization to seed the 96-wells plate. Please give some additional indications.
· Page 4, § Statistics : the last sentence of this paragraph must be removed (instruction to authors).
· Page 5 Figure 1, panel a and b: why viabilities of control (C) are different from 100% ?
· Page 5 Figure 1, panel a: How AF1a+AN1 could be different from AF1a alone (“b” on the corresponding bar) but not different from the Control (no “*”) since Control and AF1a appear strictly similar on the graph ? The last sentence page 4 could thus be modified.
· Page 5 Figure 1, panel a and b: decreased viabilities measured after exposure to AN3 or AV3 alone were the more pronounced ones despite the fact that these samples were FB2 negative or the lowest for STC. This point must be mentioned in § results and discussed then.
· Page 5, legend Figure 1 ; Page 6, legend Figure 2 ; Page 7, legend Figure 3 and Page 8, legend Figure 4: “a”- significantly different compared to AV1a, AV2a or AF3a (panel b)” and “b- significantly different compared to AF1b or AF2b (panel A) and AV1b, AV2b or AF3b (panel B)”. Since samples AF3a and AF3b were not mentioned in Table 1, I presume they have to be replaced by AV3a and AV3b in these legends.
· Page 6, Figure 2, panel a and b: towards THP-1 cells, significant reductions of viability were again measured after exposure to negative (AN3) or poor (AV3) mycotoxins samples tested alone, but also after exposure to the negative AF2 sample (free of AFB1). This point must be mentioned in § results and discussed then.
· Page 6: “all combination of Flavi and Nigri had more pronounced effect on DNA damage compared to the corresponding single treatment (Figure 3a) ». The tail intensity reported for AF1a,b+AN2 were not higher than those observed for the single treatment AN2. Thus, the comment should be modified.
· Page 10, lines 62-83: this paragraph corresponds more to a repeated description of the results than to a discussion of them. It can therefore be shunted.
· Page 10, line 92: “TNF-α is important an endogenous pyrogen » à TNF-α is an important endogenous pyrogen.
Author Response
The authors would like to thank to the reviewer for their valuable comments and suggestions!
Reviewer 1: Page 2, § 2.1. Chemicals and media : DMSO appears twice, from two different suppliers
ANSWER: Thank you. I added an explanation in the manuscript- two different grades of DMSO were used: molecular biology grade for the treatment preparation and a technical grade for dissolving the formazane in MTT assay
Reviewer 1:· Page 3, § 2.2.2. MTT Proliferation Assay : “the differentiation of THP-1 cells into macrophages was performed using PMA (40 nmol/l)”. It is not clear when this differentiation was performed. Is it directly in the 96 wells plate or in the 75 cm2 flask culture, i.e. that requires a trypsinization to seed the 96-wells plate. Please give some additional indications.
ANSWER: Thank you. The differentiation was performed directly in 96-well plates for the cytotoxicity assay (marked yellow)
Reviewer 1:· Page 4, § Statistics : the last sentence of this paragraph must be removed (instruction to authors).
ANSWER: Thank you, it is removed now.
Reviewer 1: Page 5 Figure 1, panel a and b: why viabilities of control (C) are different from 100% ?
ANSWER: Thank you for this comment- the explanation was included in the caption of the figures. Viability of the cells treated with DMSO containing medium was expressed as a percentage of viability produced in medium without DMSO. A sentence was also added in the Materials and Methods section.
Reviewer 1:· Page 5 Figure 1, panel a: How AF1a+AN1 could be different from AF1a alone (“b” on the corresponding bar) but not different from the Control (no “*”) since Control and AF1a appear strictly similar on the graph ? The last sentence page 4 could thus be modified.
ANSWER: Thank you for the comment- the mistakes on the figures were tackled and new Figure 1 is now inserted.
Reviewer 1: Page 5 Figure 1, panel a and b: decreased viabilities measured after exposure to AN3 or AV3 alone were the more pronounced ones despite the fact that these samples were FB2 negative or the lowest for STC. This point must be mentioned in § results and discussed then.
ANSWER: Thank you. A paragraph on this is included in the discussion.
Reviewer 1:· Page 5, legend Figure 1 ; Page 6, legend Figure 2 ; Page 7, legend Figure 3 and Page 8, legend Figure 4: “a”- significantly different compared to AV1a, AV2a or AF3a (panel b)” and “b- significantly different compared to AF1b or AF2b (panel A) and AV1b, AV2b or AF3b (panel B)”. Since samples AF3a and AF3b were not mentioned in Table 1, I presume they have to be replaced by AV3a and AV3b in these legends.
ANSWER: Thank you, this is now corrected.
Reviewer 1:· Page 6, Figure 2, panel a and b: towards THP-1 cells, significant reductions of viability were again measured after exposure to negative (AN3) or poor (AV3) mycotoxins samples tested alone, but also after exposure to the negative AF2 sample (free of AFB1). This point must be mentioned in § results and discussed then.
ANSWER: Thank you. This is now included in the results and discussion.
Reviewer 1:· Page 6: “all combination of Flavi and Nigri had more pronounced effect on DNA damage compared to the corresponding single treatment (Figure 3a) ». The tail intensity reported for AF1a,b+AN2 were not higher than those observed for the single treatment AN2. Thus, the comment should be modified.
ANSWER: Thank you. It is modified as suggested.
Reviewer 1:· Page 10, lines 62-83: this paragraph corresponds more to a repeated description of the results than to a discussion of them. It can therefore be shunted.
ANSWER: Thank you. Part 62-67 remained in the article and the part from 68-83 was excluded from the discussion.
Reviewer 1:· Page 10, line 92: “TNF-α is important an endogenous pyrogen » à TNF-α is an important endogenous pyrogen.
ANSWER: Thank you, it was corrected.
Reviewer 2 Report
Jakšić et al evaluate the interspecies and interseries difference in toxigenicity of aspergilli flavi, nigri and versicolores and their combined effect in vitro using A549 and THP-1 macrophages. This study does provide interesting aspect of toxicity however is real application and relation to actual human exposure is unclear.
· How compatible between the concentration tested in the manuscript and the estimate concentration of mycotoxin that is inhaled and retain in the respiratory system (PMID: 15162841)? The lab culture provide optimal condition for mycotoxin production, which is generally higher than the actual indoor exposure and toxic effect observed in human, it is difficult to ascertain the true effect and health risk in human subject
· The production of mycotoxin are highly variable depending on the substrate, host and environment, are this variable consider and how this will affect the cytotoxic, genotoxic, and pro-inflammatory properties should be discuss
· The limitation of the study should be stated and discuss.
Author Response
The authors would like to thank to the reviewer for their valuable comments and suggestions!
Reviewer 2: Jakšić et al evaluate the interspecies and interseries difference in toxigenicity of aspergilli flavi, nigri and versicolores and their combined effect in vitro using A549 and THP-1 macrophages. This study does provide interesting aspect of toxicity however is real application and relation to actual human exposure is unclear.
ANSWER: Thank you for your comment, it is very much appreciated! Some extra paragraphs are now included in the discussion so we believe it better explains the significance of the results.
Reviewer 2: · How compatible between the concentration tested in the manuscript and the estimate concentration of mycotoxin that is inhaled and retain in the respiratory system (PMID: 15162841)? The lab culture provide optimal condition for mycotoxin production, which is generally higher than the actual indoor exposure and toxic effect observed in human, it is difficult to ascertain the true effect and health risk in human subject
ANSWER: Thank you very much for your comment. The discussion part is now supplemented with the data on significance of the investigated concentrations. The authors agree on the point stated, of course these models are not ideal and are only approximative. Actual exposere to indoor fungal metabolites may be best evaluated through inhalable dust, and available published data identified many Aspergillus metabolites in such samples in the quantities comparable to this study.
On the other hand, a special aim of this research was point to the differences in the toxic potential in regard to the combination treatment which differs from the treatment with a single extract. The interpretation of the results may go in many ways. For this purpose we decided to point to the concept of the species which not always considers the chemical composition. Another aspect is the role of other metabolites in the toxicity besides the recognised mycotoxins and related impact of the complexity of metabolites on the toxicity endpoint.
Reviewer 2: The production of mycotoxin are highly variable depending on the substrate, host and environment, are this variable consider and how this will affect the cytotoxic, genotoxic, and pro-inflammatory properties should be discuss
ANSWER: Thank you for your comment, it is much appriciated and included in the discussion.
Reviewer 2: · The limitation of the study should be stated and discuss.
ANSWER: Thank you for your comment, it is much appriciated and included in the discussion.